# Secreted antigen A peptidoglycan hydrolase is essential for *Enterococcus faecium* cell separation and priming of immune checkpoint inhibitor therapy

**Steven Klupt[1†], Kyong Tkhe Fam[1†], Xing Zhang[1†], Pavan Kumar Chodisetti[1], Abeera Mehmood[1], Tumara Boyd[2], Danielle Grotjahn[2], Donghyun Park[2], Howard C Hang[1,3]***

[1]Department of Immunology and Microbiology, Scripps Research, La Jolla, United States; [2]Department of Integrative Structural & Computational Biology, Scripps Research, La Jolla, United States; [3]Department of Chemistry, Scripps Research, La Jolla, United States

**\*For correspondence:**
hhang@scripps.edu

[†]These authors contributed equally to this work

**Abstract** *Enterococcus faecium* is a microbiota species in humans that can modulate host immunity (Griffin and Hang, 2022), but has also acquired antibiotic resistance and is a major cause of hospital-associated infections (Van Tyne and Gilmore, 2014). Notably, diverse strains of *E. faecium* produce SagA, a highly conserved peptidoglycan hydrolase that is sufficient to promote intestinal immunity (Rangan et al., 2016; Pedicord et al., 2016; Kim et al., 2019) and immune checkpoint inhibitor antitumor activity (Griffin et al., 2021). However, the functions of SagA in *E. faecium* were unknown. Here, we report that deletion of *sagA* impaired *E. faecium* growth and resulted in bulged and clustered enterococci due to defective peptidoglycan cleavage and cell separation. Moreover, Δ*sagA* showed increased antibiotic sensitivity, yielded lower levels of active muropeptides, displayed reduced activation of the peptidoglycan pattern-recognition receptor NOD2, and failed to promote cancer immunotherapy. Importantly, the plasmid-based expression of SagA, but not its catalytically inactive mutant, restored Δ*sagA* growth, production of active muropeptides, and NOD2 activation. SagA is, therefore, essential for *E. faecium* growth, stress resistance, and activation of host immunity.

## eLife assessment

The authors build upon prior data implicating the secreted peptidoglycan hydrolase SagA produced by *Enterococcus faecium* in immunotherapy. Leveraging new strains with sagA deletion/complementation constructs, the investigators reveal that sagA is non-essential, with sagA deletion leading to a marked growth defect due to impaired cell division, and sagA being necessary for the immunogenic and anti-tumor effects of *E. faecium*. In aggregate, the study utilizes **compelling** methods to provide both **fundamental** new insights into *E. faecium* biology and host interactions and a proof-of-concept for identifying the bacterial effectors of immunotherapy response.

## Introduction

*Enterococcus* is a genus of Gram-positive bacteria that is composed of more than seventy different species found in diverse environments, both free-living and in relationships with various animals (*Lebreton et al., 2017*). *Enterococcus faecium* strains have been isolated from humans and reported to have both beneficial and pathogenic properties (*Van Tyne and Gilmore, 2014*). Notably,

antibiotic-resistant strains of *E. faecium*, particularly vancomycin-resistant *E. faecium* (VREfm), have emerged as major causes of healthcare-associated infections (*Van Tyne and Gilmore, 2014*; *Fiore et al., 2019*; *García-Solache and Rice, 2019*), and have been correlated with graft-versus-host disease (GVHD) and increased mortality in allogeneic hematopoietic cell transplantation patients (*Stein-Thoeringer et al., 2019*). *E. faecium* has also been recovered from ulcerative colitis and Crohn's disease patients and has been shown to exacerbate intestinal inflammation and colitis in mouse models of inflammatory bowel disease (IBD) *Barnett et al., 2010*; *Seishima et al., 2019*. However, commensal strains of *E. faecium* have also been reported to enhance intestinal immunity in animal models and have been developed as probiotics (*Hanchi et al., 2018*). Furthermore, microbiota analysis has shown that *E. faecium* was enriched in immune checkpoint inhibitor (ICI) (*Gopalakrishnan et al., 2018*; *Routy et al., 2018*; *Matson et al., 2018*) and chimeric antigen receptor (CAR) T-cell therapy-responsive patients (*Smith et al., 2022*). These studies highlight the potential pathogenic and beneficial features of *E. faecium*.

Our laboratory previously investigated the beneficial effects of *E. faecium* on host physiology (*Griffin and Hang, 2022*). We demonstrated that secreted antigen A (SagA), a highly conserved NlpC/P60 peptidoglycan hydrolase in *E. faecium*, was sufficient to confer protection against enteric infections in both *Caenorhabditis elegans* and mice (*Rangan et al., 2016*). In mice, *E. faecium* and an *E. faecalis* strain engineered to express SagA both up-regulated expression of mucins and antimicrobial peptides, resulting in improved intestinal barrier function as well as tolerance to *Salmonella enterica* serovar Typhimurium and *Clostridioides difficile* pathogenesis (*Pedicord et al., 2016*; *Kim et al., 2019*). In contrast, wild-type *E. faecalis*, which does not express SagA, did not exhibit these effects. We then determined the X-ray crystal structure of the SagA NlpC/P60 hydrolase domain, and demonstrated that this hydrolase preferentially cleaves crosslinked peptidoglycan fragments into smaller muropeptides (such as GlcNAc-MDP), which more effectively activate the peptidoglycan pattern recognition receptor NOD2 (nucleotide-binding oligomerization domain-containing protein 2) in mammalian cells (*Kim et al., 2019*). Importantly, NOD2 was shown to be required for *E. faecium* stimulation of intestinal immunity and tolerance to infection in vivo (*Pedicord et al., 2016*; *Kim et al., 2019*). Moreover, *E. faecium* and SagA were sufficient to protect mice against dextran sodium sulfate-induced colitis that required the expression of NOD2 in myeloid cells (*Jang et al., 2023*).

The discovery of *E. faecium* among the microbiota of cancer immunotherapy-responsive patients then motivated our analysis of *Enterococcus* species and SagA in mouse tumor models. Indeed, diverse strains of *E. faecium* that express SagA, but none of the non-SagA-expressing *E. faecalis* strains evaluated, were sufficient to promote ICI (anti-PD-L1, anti-PD-1, anti-CTLA-4) anti-tumor activity against different cancer types in mouse models (*Griffin et al., 2021*). Furthermore, other *Enterococcus* species including *E. durans*, *E. hirae*, and *E. mundtii*, which each have SagA orthologs with greater than 80% protein similarity to that of *E. faecium*, were sufficient to enhance anti-PD-L1 antitumor activity (*Griffin et al., 2021*). The SagA orthologs from these other *Enterococcus* species were expressed and secreted at similar levels to *E. faecium* SagA and showed similar peptidoglycan hydrolase activity in vitro (*Griffin et al., 2021*). Importantly, heterologous expression of SagA in inactive bacterial species (*E. faecalis* and *Lactococcus lactis*) was sufficient to promote anti-PD-L1 antitumor activity and required NOD2 in vivo (*Griffin et al., 2021*). Collectively, these studies demonstrated that *Enterococcus* peptidoglycan remodeling by SagA is sufficient to enhance intestinal immunity against infection and promote cancer immunotherapy. However, the endogenous functions of SagA in *E. faecium* microbiology and modulation of host immunity were not investigated previously, as *sagA* was believed to be an essential gene in *E. faecium* (strain TX1330), as reported by Murray and coworkers (*Teng et al., 2003*). Nevertheless, our collaborative studies with the Duerkop laboratory have indicated that phage-resistant strains of *E. faecium* (strain Com12) containing catalytically inactivating point mutations in *sagA* were still viable (*Canfield et al., 2023*), suggesting that SagA may not be essential and could be functionally evaluated in *E. faecium*.

## Results

Here, we report the generation of a Δ*sagA* strain of *E. faecium* (Com15). This strain, generated using RecT-mediated recombineering methods developed by our laboratory (*Chen et al., 2021*), exhibited significantly impaired growth (*Figure 1a*) and sedimentation in liquid culture (*Figure 1—figure supplement 1b*). We then performed whole-genome sequencing to assess the fidelity of the Δ*sagA*

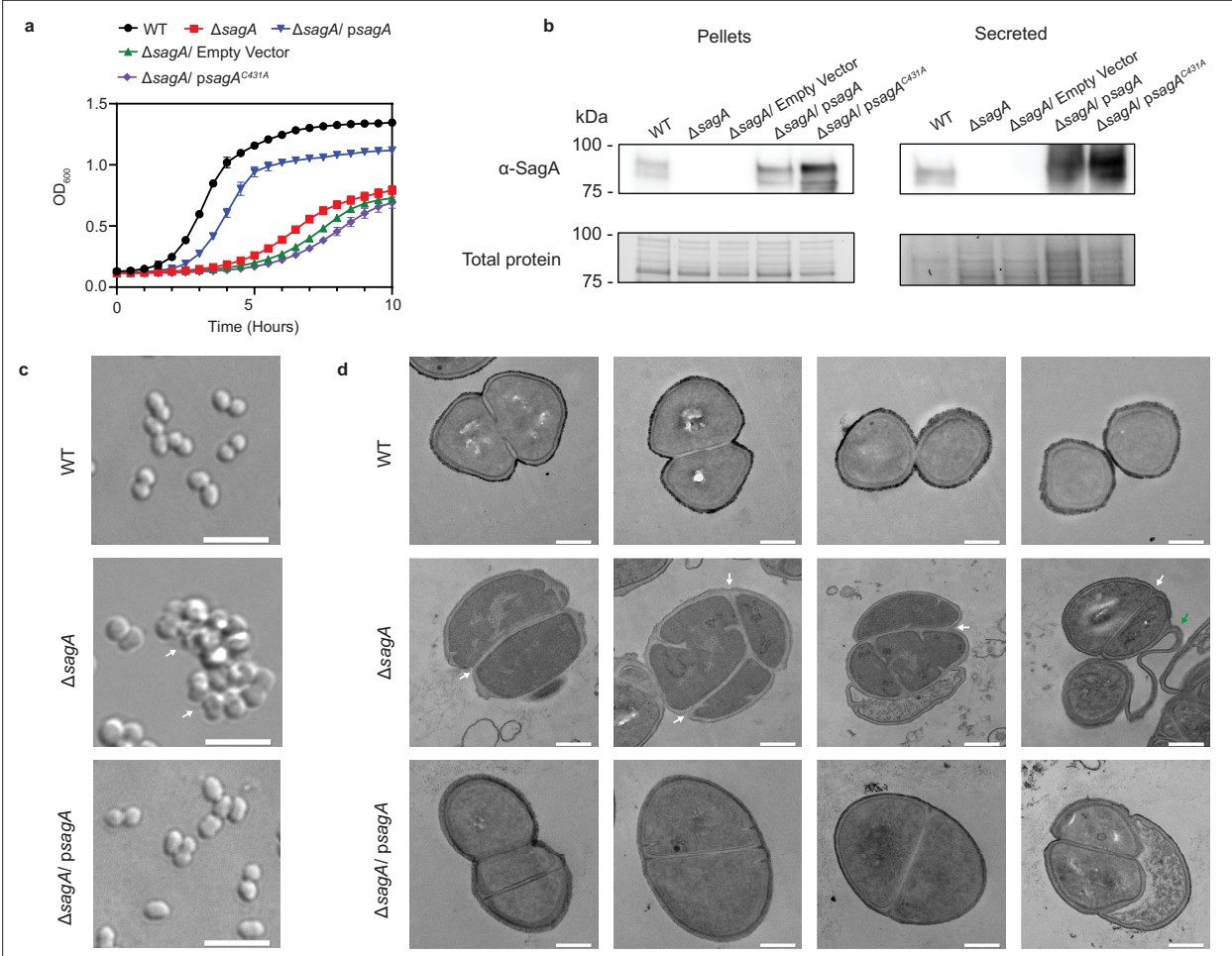

**Figure 1.** Growth and morphology phenotypes of wild-type, ΔsagA, and psagA complemented *E. faecium* Com15 strains. (**a**) Growth curves of *E. faecium* WT, ΔsagA, and complementation strains with functional and nonfunctional sagA genes (n=3). Data are presented as mean value ± standard deviation. (**b**) α-SagA western blot of *E. faecium* WT, ΔsagA, and complementation strains with functional and nonfunctional sagA genes, on both whole cell lysate (pellets) and total secreted proteins (secreted). Bottom panel shows total protein loading visualized by Stain-free imaging and serves as protein loading control. (**c**) Differential interference contrast (DIC) microscopy of *E. faecium* WT, ΔsagA, and ΔsagA/psagA complemented strains. White arrows point to cell clusters. Scale bar = 5 µm. (**d**) Transmission electron microscopy (TEM) of *E. faecium* WT, ΔsagA, and ΔsagA/psagA complemented strains. White arrows point to failed cell separation in ΔsagA. Green arrow points to undegraded peptidoglycan in ΔsagA. Scale bar = 0.2 µm.

The online version of this article includes the following source data and figure supplement(s) for figure 1:

**Source data 1.** Excel file containing numeric data used to generate *Figure 1a*.

**Source data 2.** Uncropped western blots and gels (total protein) used to generate *Figure 1b*.

**Source data 3.** Uncropped images used to generate *Figure 1c*.

**Source data 4.** Uncropped images used to generate *Figure 1d*.

**Source data 5.** Validation of sagA deletion and image of bacterial strain growth.

**Figure supplement 1.** Analysis of *E.faecium* ΔsagA deletion and complementation studies.

**Figure supplement 1—source data 1.** Uncropped gel used to generate *Figure 1—figure supplement 1a* and uncropped photo used to generate *Figure 1—figure supplement 1b*.

**Figure supplement 2.** Antibiotic sensitivity of Δ*sagA*.

**Figure supplement 2—source data 1.** Uncropped images used to generate *Figure 1—figure supplement 2a*.

**Figure supplement 2—source data 2.** Excel file containing numeric data used to generate *Figure 1—figure supplement 2b*.

strain's genome to that of wild-type. While ΔsagA contained 14 mutations, most were outside of open-reading frames (ORFs) or were in genes that are unrelated to peptidoglycan metabolism (*Supplementary file 1*). One notable mutation, however, was in the predicted glycosidase EFWG_00994. This mutation, L200F, alters a leucine residue that is conserved among GH73 family members and the other putative Com15 glucosaminidases (*Supplementary file 1*). To confirm that the ΔsagA growth defect was truly SagA-dependent and was not an outcome of suppressor mutations, we generated a complementation strain (ΔsagA/ psagA) that contained a plasmid expressing sagA under control of its native promoter. Western blot analysis with α-SagA polyclonal serum confirmed the deletion and complementation of SagA (*Figure 1b*). Differential interference contrast (DIC) microscopy revealed that ΔsagA cells form irregular clusters, as opposed to the typical wild-type morphology of diplococci or short chains (*Figure 1c*). This cell morphology suggested that dividing ΔsagA cells may be unable to separate from one another during binary fission. Indeed, transmission electron microscopy (TEM) of chemically fixed samples confirmed that ΔsagA cells have defects in cell separation (*Figure 1d*). Division septa could form and mature in these cells; however, daughter cells failed to separate, resulting in clusters of unseparated cells. Cells on the periphery of these clusters were highly strained, and in some cases appeared to lyse and leave behind strands of undegraded peptidoglycan (*Figure 1d*). Importantly, the complementation largely restored the log-phase growth rate and stationary-phase OD of the ΔsagA strain to those of wild-type (*Figure 1a*). It also reversed the liquid-culture sedimentation phenotype (*Figure 1—figure supplement 1b*) and reduced the cell clustering (*Figure 1c*) and defective septal separation (*Figure 1d*).

To investigate the significance of peptidoglycan hydrolase activity in SagA, we generated a catalytically inactive point mutant (C431A) in our sagA complementation plasmid. Consistent with our previous biochemical results on recombinant SagA NlpC/P60 domain activity (*Kim et al., 2019*; *Espinosa et al., 2020*), the C431A mutant was unable to complement the sagA deletion (*Figure 1a–d*), despite Western blot analysis confirming that the mutant SagA was expressed and secreted properly (*Figure 1b*). Together, these results demonstrated that catalytically active SagA is required for proper E. faecium growth.

Based on the growth defect of ΔsagA and the previously reported antibiotic sensitization effect of catalytically inactivating point mutants in the SagA of E. faecium strain Com12 (*Canfield et al., 2023*), we chose to investigate the impact of the sagA deletion on the effectiveness of various cell wall-acting antibiotics. Using minimum inhibitory concentration (MIC) test strips of common antibiotics (vancomycin, linezolid, daptomycin, tigecycline, telavancin, fosfomycin, ampicillin, ceftriaxone, and imipenem), we observed increased sensitivity to the β-lactams (ampicillin, ceftriaxone, and imipenem), moderately increased sensitivity to daptomycin, tigecycline, fosfomycin, and linezolid and no change in sensitivity to telavancin and vancomycin (*Figure 1—figure supplement 2a*, *Supplementary file 2*). MIC analysis of ampicillin in liquid culture, further demonstrated that ΔsagA has increased susceptibility to β-lactam antibiotics, which was abrogated with psagA expression (*Figure 1—figure supplement 2b*).

To further investigate the growth defect and increased β-lactam antibiotic sensitivity of ΔsagA, we performed cryo-electron tomography (cryo-ET) on frozen-hydrated samples to quantify factors such as peptidoglycan thickness, septum thickness, and placement of divisome components at a higher resolution (*Figure 2—figure supplement 1a and b*). There was no statistically significant change in peptidoglycan thickness and septum thickness in ΔsagA and ΔsagA/ psagA cells compared to wild-type cells (*Figure 2—figure supplement 1c and d*). Similar to previous studies of Bacillus subtilis (*Khanna et al., 2021*), the divisome machinery was directly observable as concentric rings in cross-sectional images generated by cryo-ET in E. faecium (*Figure 2—figure supplement 2a–d*). We quantified the distances from these rings in order to assess potential differences in divisome placement (*Figure 2—figure supplement 2e*). The distance between the septal membrane and the distal ring of the divisome showed a slight increase, but we did not observe any statistical difference between the distance from the septal membrane to the proximal ring and the ratio between the proximal and distal rings in wild-type, ΔsagA, and complementation (*Figure 2—figure supplement 1e*, *Figure 2—figure supplement 2f and g*). However, we observed that the placement and projection angle of growing septa were significantly altered in ΔsagA but were mostly restored by complementation (*Figure 2a–f*). Both room temperature TEM and cryo-ET revealed that sagA complementation largely restored the wild-type cell morphology of ΔsagA to that of wild-type (*Figure 2*, *Figure 2—figure supplements 1*

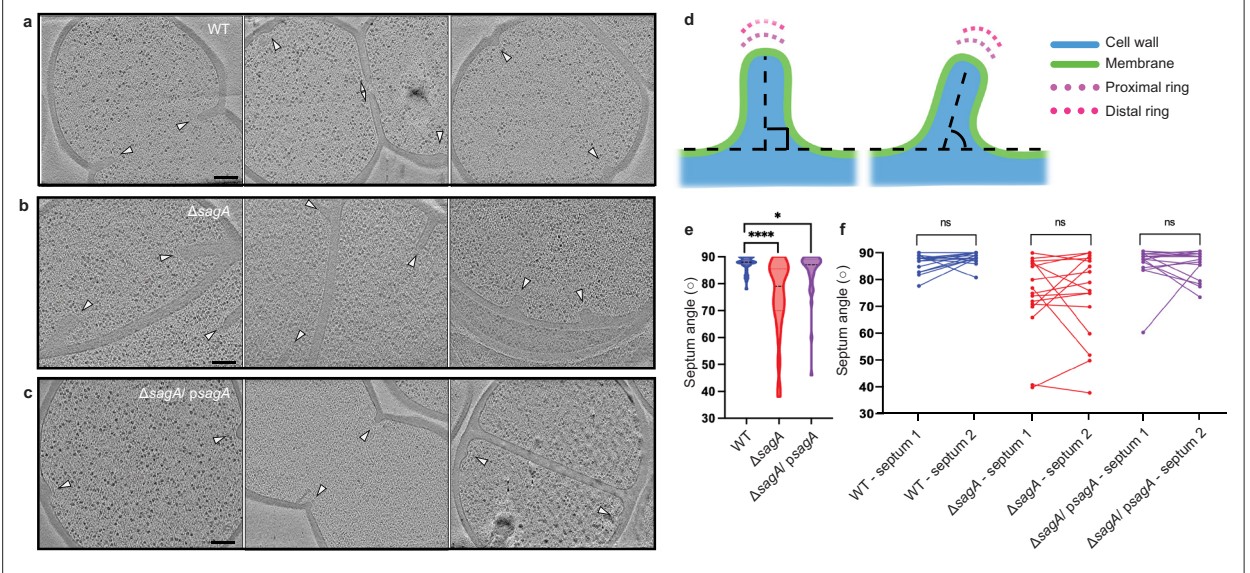

**Figure 2.** Deletion of *sagA* changes the angle of growing septum. (**a-c**) Representative tomographic slices (n=3) of *E. faecium* strains: *E. faecium* wild-type (WT) (**a**), Δ*sagA* (**b**), and Δ*sagA*/p*sagA* (**c**). Cell division septa are indicated by white arrows. Scale bar = 100 nm. (**d**) A diagram indicating how septum angle measurements were collected. Acute angles are recorded for further analysis. (**e**) Comparison of septum angle. The violin plot displays the distribution of septum angle, with *E. faecium* WT (n=40) shown in blue, Δ*sagA* (n=49) shown in red, and Δ*sagA*/p*sagA* (n=37) shown in magenta. Black dotted lines represent median (*E. faecium* WT: 88°, Δ*sagA*: 79°, Δ*sagA*/ p*sagA*: 87°) while the colored dotted lines represent quartiles. Welch's *t*-test was used to calculate statistical significance. *p<0.05; ****p<0.0001. (**f**), Pairwise comparison of septum angle in opposing septa. The paired plot displays the distribution of septum angle, with *E. faecium* WT (n=18) shown in blue, Δ*sagA* (n=16) shown in red, and Δ*sagA*/ p*sagA* (n=16) shown in magenta. Two septum angles from opposing septa are linked with straight lines. Paired *t*-test was used to calculate statistical significance. ns, p≥0.05.

The online version of this article includes the following source data and figure supplement(s) for figure 2:

**Source data 1.** Excel file containing numeric data to generate *Figure 2e–f*.

**Figure supplement 1.** *E.faecium* Δ*sagA* exhibits no defects in cell wall and septum thickness.

**Figure supplement 1—source data 1.** Excel file containing numeric data to generate *Figure 2—figure supplement 1c–e*.

**Figure supplement 2.** Deletion of *sagA* alters the position of cell division.

**Figure supplement 2—source data 1.** Excel file containing numeric data used to generate *Figure 2—figure supplement 2f–g*.

*and 2*). The complemented cells separate from one another and mostly take the form of diplococci. These results suggest that the ultrastructure of the peptidoglycan cell wall is unchanged as a result of *sagA* deletion, but peptidoglycan cleavage, divisome angle, and septal resolution are impaired.

We then investigated Δ*sagA* peptidoglycan composition and activation of NOD2 immune signaling. To evaluate peptidoglycan composition, *E. faecium* sacculi were isolated, subjected to mutanolysin digestion, analyzed by liquid chromatography mass spectrometry (*Figure 3—figure supplement 1*) and quantified (*Figure 3a and b*), as previously described (*Kim et al., 2019*). The Δ*sagA* strain showed decreased levels of small muropeptides (peaks 2, 3, and 7 with green asterisk) compared to *E. faecium* Com15, which was restored in Δ*sagA*/ p*sagA* (*Figure 3a*). Moreover, we observed increased amounts of crosslinked peptidoglycan fragments (peaks 13 and 14 with purple asterisk) (*Figure 3a*). Notably, GlcNAc-MDP, which we previously demonstrated more effectively activates NOD2 (*Kim et al., 2019*), was significantly decreased in Δ*sagA* (*Figure 3b*). We next evaluated live bacterial cultures with mammalian cells to determine their ability to activate the peptidoglycan pattern recognition receptor NOD2. Indeed, our analysis of these bacterial strains using HEK-Blue NF-κB reporter cells demonstrated that Δ*sagA* exhibits significantly decreased NOD2 activation compared to *E. faecium* WT. This activation was restored in Δ*sagA*/ p*sagA* (*Figure 3c*). For these assays, we used comparable numbers of bacteria to account for defects in Δ*sagA* growth, which was confirmed by colony-forming unit analysis of bacteria per well (*Figure 3d*). These results were consistent with our in vitro analysis of recombinant SagA with purified peptidoglycan fragments, which showed that the generation of small muropeptides (GlcNAc-MDP) more potently activated NOD2 compared to crosslinked muropeptides (*Kim et al., 2019*). Our results also demonstrated that while many enzymes are required for the

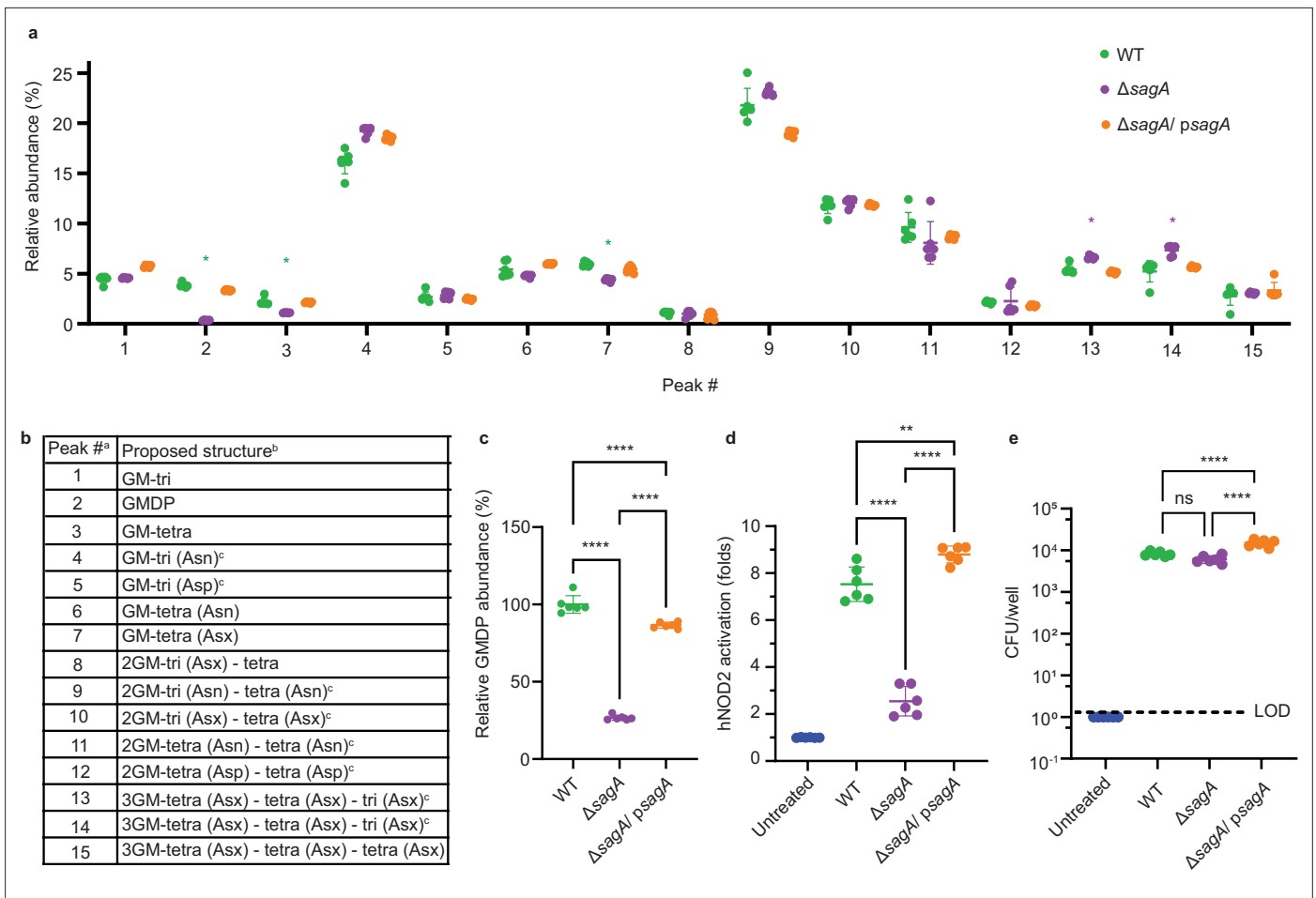

**Figure 3.** Peptidoglycan profile and NOD2 activation of *E.faecium* Δ*sagA*. (**a**) Relative abundance of muropeptides isolated from mutanolysin-digested sacculi of *E. faecium* strains and analyzed by LC-MS (n=6). Green asterisks highlight changes in abundance of small muropeptides, purple asterisks highlight changes in abundance of crosslinked peptidoglycan fragments. Numbers correspond to different muropeptides from LC-MS analysis listed in legend. (**b**) Composition of muropeptides from *E. faecium* sacculi. [a] Peak numbers refer to (**a**). [b] GM, disaccharide (GlcNAc-MurNAc); 2 GM, disaccharide-disaccharide (GlcNAc-MurNAc-GlcNAc-MurNAc); 3 GM, disaccharide-disaccharide-disaccharide (GlcNAc-MurNAc-GlcNAc-MurNAc-GlcNAc-MurNAc); GM-Tri, disaccharide tripeptide (L-Ala-D-iGln-L-Lys); GM-Tetra, disaccharide tetrapeptide (L-Ala-D-iGln-L-Lys-D-Ala); GM-Penta, disaccharide pentapeptide (L-Ala-D-iGln-L-Lys-D-Ala -D-Ala). [c] The assignment of the amide and the hydroxyl functions to either peptide stem is arbitrary. (**c**) Relative abundance of GMDP relative to wild-type (WT) from LC-MS chromatograms (n=6). (**d**), NF-$\kappa$B responses of HEK-Blue hNOD2 cells to live *E. faecium* strains (MOI = 1, n=6). NOD2 activation is expressed as fold change relative to untreated control. (**e**) Colony forming units (CFU) of *E. faecium* strains (MOI = 1) internalized in HEK-Blue hNOD2 cells (n=6). Dashed line indicates Limit of Detection (LOD). Data for (**a, c–e**) represent mean value ± standard deviation and analyzed with one-way ANOVA and Tukey's multiple comparison post hoc test. *p≤0.05; **p≤0.01; ***p≤0.005; ****p≤0.001; ns, not significant.

The online version of this article includes the following source data and figure supplement(s) for figure 3:

**Source data 1.** Excel file containing numeric data used to generate *Figure 3a and c–e*.

**Figure supplement 1.** Peptidoglycan profile of *E. faecium* Δ*sagA* by LC-MS.

**Figure supplement 1—source data 1.** Excel file containing numeric data used to generate *Figure 3—figure supplement 1a*.

biosynthesis and remodeling of peptidoglycan in *E. faecium*, SagA is essential for generating NOD2-activating muropeptides ex vivo.

We next investigated if SagA is crucial for *E. faecium* colonization and immune modulation in vivo using well-established mouse models of ICI cancer immunotherapy (*Figure 4a*). We had previously demonstrated that heterologous expression of SagA was sufficient to promote *E. faecalis* and *L. lactis* ICI antitumor activity and required *Nod2* expression in mice (*Griffin et al., 2021*). However, activity of wild-type *E. faecium* in *Nod2*[-/-] mice was not determined. Indeed, oral administration of *E. faecium* to microbiota-depleted/antibiotic-treated C57BL/6 mice promoted anti-PD-1 antitumor activity against

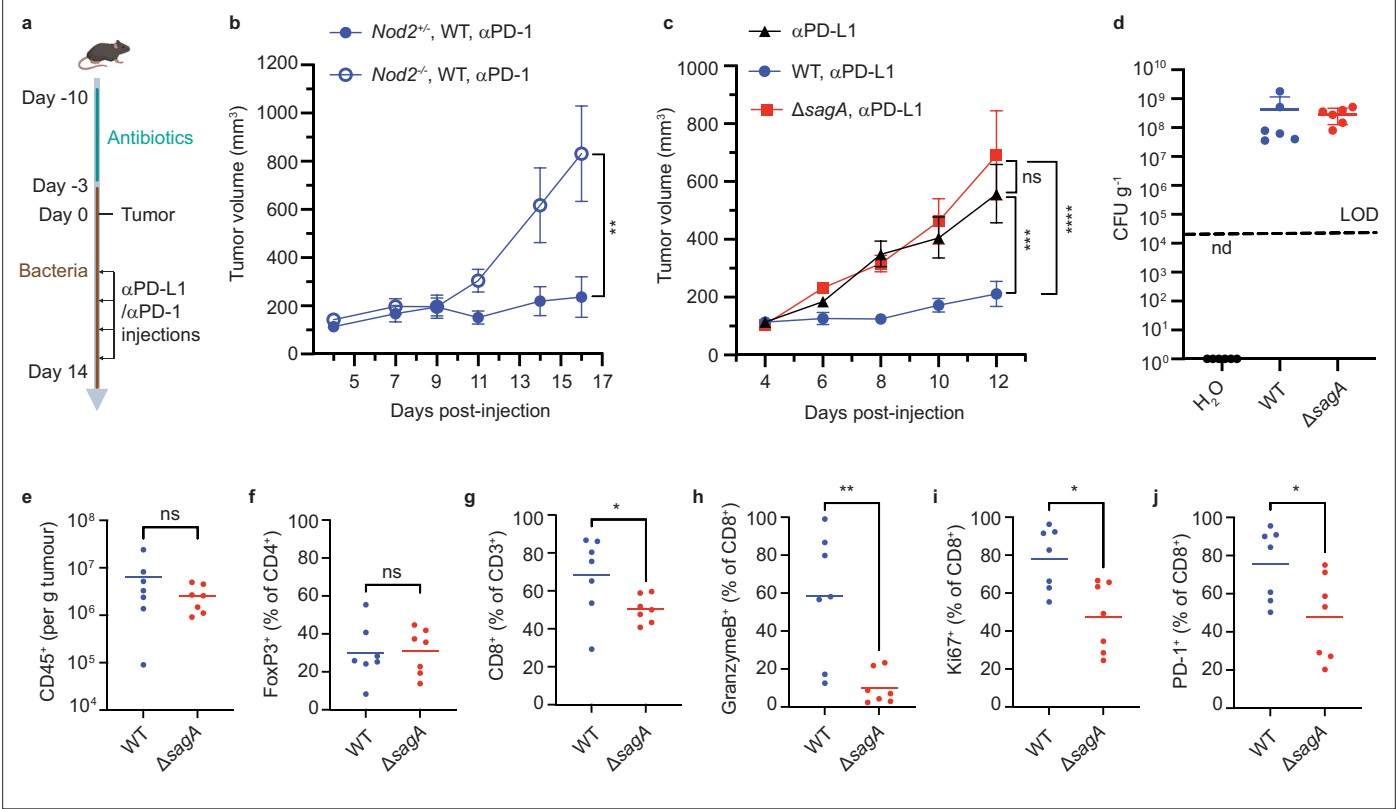

**Figure 4.** Immune checkpoint inhibitor antitumor activity and tumor immune profile of *E.faecium* Δ*sagA* colonized mice. (**a**) Schematic of tumor growth experiment: mice were provided water containing antibiotics for one week and started drinking bacteria three days before tumor implantation. Once the tumor reaches ~100 mm³, the measurement starts, and two days after treated with anti-PD-1 (MC38) or anti-PD-L1 (B16F10) every other day. (**b**) MC-38 tumor growth in *Nod2*⁺/⁻ or *Nod2*⁻/⁻ mice that were colonized with *E. faecium* WT and treated with anti-PD-1 starting at day 7. +/-=10 for *Nod2*⁺/⁻ +/-, n=6 for *Nod2*⁻/⁻ mice. (**c**) B16F10 tumor growth in C57BL/6 mice that were colonized with *E. faecium* wild-type (WT) or Δ*sagA* and treated with anti-PD-L1 starting at day 6. No bacterial colonization group as a control (black). n=7–8 mice per group. (**d**) Fecal colony forming units (CFU) analysis of *E. faecium* on HiCrome *Enterococcus faecium* agar plates from (**c**) at day 6. n=6 per group. Each dot represents one mouse. The line indicates the limit of detection (LOD, 4000 CFU g⁻¹). Nd, not detected. Data represent means ± 95% confidence interval. (**e–j**) Quantification of tumor infiltrating CD45⁺ cells (**e**), FoxP3⁺ cells (**f**), CD3⁺ CD8⁺ cells (**g**), GranzymeB⁺ CD8⁺ T cells (**h**), Ki67⁺ CD8⁺ T cells (**i**), and PD-1⁺ CD8⁺ T cells (**j**). For (**f, h-j**), fluorescence minus one (FMO) control was used to define gates. n=7 mice per group. Data for (**b**) and (**c**) represent mean ± SEM and were analyzed using a mixed effects model with Tukey's multiple comparisons post hoc test. Data for (**e–j**) represent mean ± SEM and were analyzed by the Mann-Whitney U (one-tail) test. *p<0.05, **p<0.01, ***p<0.001, ****p<0.0001; ns, not significant.

The online version of this article includes the following source data and figure supplement(s) for figure 4:

**Source data 1.** Excel file containing numeric data used to generate *Figure 4b–j*.

**Figure supplement 1.** Gating strategy used for flow cytometry analysis of tumor infiltrating lymphocytes (TILs).

**Figure supplement 1—source data 1.** Excel file containing numeric data used to generate *Figure 4—figure supplement 1*.

MC-38 tumor growth in *Nod2*⁺/⁻, but failed to do so in *Nod2*⁻/⁻ mice (*Figure 4b*). To determine if SagA is required for *E. faecium* function in vivo, antibiotic-treated mice were colonized with *E. faecium* Com15 or Δ*sagA*. Notably, the antitumor activity of *E. faecium* was abolished in Δ*sagA* strain (*Figure 4c*) even though both bacterial strains colonized mice at comparable levels, as judged by fecal *E. faecium* levels (*Figure 4d*). We then performed immune profiling of tumor infiltrating lymphocytes (TILs) using flow cytometry (*Figure 4—figure supplement 1*). Compared to *E. faecium* Com15 colonized mice, Δ*sagA* colonized mice showed no difference in the total amount of the tumor-infiltrating CD45⁺ cells (*Figure 4e*) or CD4⁺ FoxP3+ regulatory T cells (*Figure 4f*), but resulted in fewer CD3⁺ CD8⁺ T cells (*Figure 4g*). Furthermore, granzymeB⁺ CD8⁺ T cells, Ki67⁺ CD8⁺ T cells, and PD-1⁺ CD8⁺ T cells levels were also lower in Δ*sagA* colonized mice (*Figure 4h–j*). We evaluated the activity Δ*sagA*/p*sagA* strain in these experiments (data not shown), but the results were inconclusive likely due to instability of p*sagA* plasmid in *E. faecium* in vivo. We note that by continuous oral administration in the drinking

water, live *E. faecium* and soluble muropeptides that are released into the media during bacterial growth may both contribute to NOD2 activation in vivo. Nonetheless, these results demonstrate SagA is not essential for *E. faecium* colonization, but is required for promoting the ICI antitumor activity through NOD2 in vivo.

## Discussion

*E. faecium* is a prominent microbiota species and pathogen in animals and humans. While *E. faecium* identification and dominance in fecal microbiota have been correlated with health and disease outcomes, the underlying mechanisms have only begun to emerge. Our previous gain-of-function studies with recombinant protein and engineered bacterial strains demonstrated that SagA, a unique secreted peptidoglycan hydrolase that is highly conserved in *E. faecium* strains and other *Enterococcus* species, but not *E. faecalis*, is sufficient to activate NOD2 and promote host immunity in vivo (*Rangan et al., 2016*; *Pedicord et al., 2016*; *Kim et al., 2019*; *Griffin et al., 2021*). However, the endogenous functions of SagA in *E. faecium* were unknown. Based on the results described above, we now show that SagA peptidoglycan hydrolase activity is not required for *E. faecium* viability, but is essential for proper growth and specifically for septal separation following cell division (*Figures 1 and 2*). These results are consistent with key peptidoglycan hydrolases in other bacterial species, such as AmiA/B in *Escherichia coli* (*Yang et al., 2011*), CwlO/LytE in *Bacillus subtilis* (*Meisner et al., 2013*; *Wilson et al., 2023*), PcsB in *Streptococcus pneumoni*ae (*Sham et al., 2011*) and RipC in *Mycobacterium tuberculosis* (*Mavrici et al., 2014*). Although the mechanisms of regulation may differ amongst peptidoglycan hydrolases and remain to be determined for SagA, the *sagA* promoter contains sequence motifs that may be regulated by WalRK two-component systems (*Brogan and Rudner, 2023*) and the SagA protein contains a N-terminal coil-coil domain that may interact with *E. faecium* divisome components, which warrants further investigation in the future. Notably, the deletion of *sagA* in *E. faecium* Com15 also renders this strain more susceptible to cell wall-acting antibiotics (*Figure 1—figure supplement 2*), akin to inactive SagA variants in *E. faecium* Com12 (*Canfield et al., 2023*), suggesting SagA and its related peptidoglycan hydrolases may be potential antibacterial targets to use in combination with existing antibiotics. Interestingly, clade B strains of *E. faecium* including Com12 and Com15 have been suggested to be reclassified as *Enterococcus lactis* based on genomic and phenotypic similarity (*Belloso Daza et al., 2022*; *Belloso Daza et al., 2021*). Nonetheless, SagA orthologs have greater than 90 percent protein sequence identity within the C-terminal NlpC/P60 hydrolase domain of ICI therapy-promoting *Enterococcus* species (*E. faecium-clade A and B, E. durans, E. hirae* and *E. mundtii*) (*Griffin et al., 2021*). Notably, all the *Enterococcus* strains and species that express active

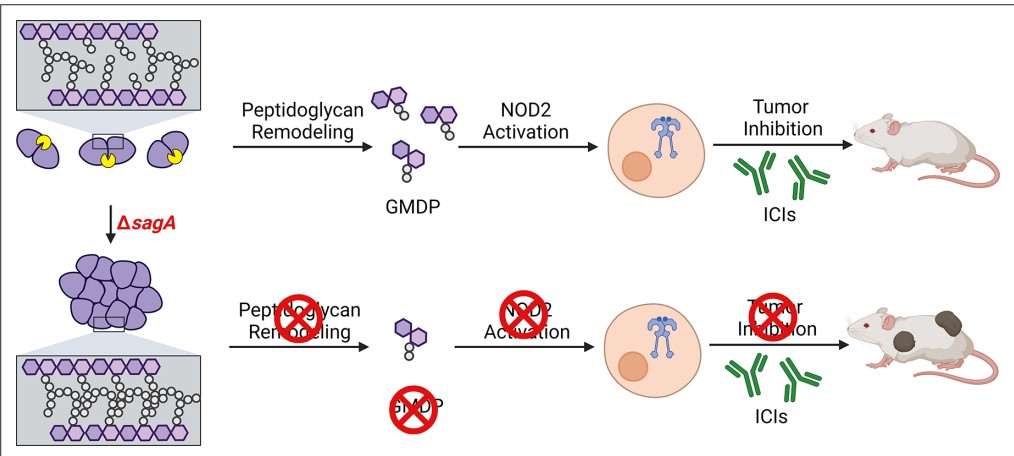

**Figure 5.** Summary of SagA function in *E. faecium* and impact on host immunity during ICI cancer therapy. Deletion of *sagA* impairs peptidoglycan remodeling and cell separation in *E. faecium* and limits the activation of NOD2 in mammalian cells to promote immune checkpoint inhibitor cancer therapy in mouse models. Created with BioRender.com.

SagA orthologs that we have analyzed promote ICI cancer immunotherapy in mouse models (*Griffin et al., 2021*).

Beyond intrinsic functions in *E. faecium* microbiology, we have also demonstrated for the first time that SagA is essential for the activation of host immunity. Notably, we showed that ΔsagA does not generate significant amounts of non-crosslinked muropeptides (i.e. GlcNAc-MDP) that are sensed by the peptidoglycan pattern recognition receptor NOD2 (*Figure 3*). Indeed, ΔsagA failed to activate NOD2 ex vivo (*Figure 3*) and promote ICI antitumor activity in mouse models compared to wild-type *E. faecium* (*Figure 4*). Collectively, our results provide important mechanistic insight on SagA in peptidoglycan remodeling for bacterial cell separation and reveal an essential feature of *E. faecium* (*Figure 5*) and other related *Enterococcus* species that are associated with human health, disease, and response to therapy.

## Materials and methods

### Bacterial growth

All cultures of *E. faecium* (*Supplementary file 3*) and *E. coli* were grown in a shaking incubator at 37 °C and 220 RPM with relevant antibiotics. *E. faecium* was grown in Brain Heart Infusion (BHI) broth or on BHI agar supplemented as needed with 10 μg/mL chloramphenicol or 50 μg/mL erythromycin. *E. coli* was grown in Luria-Bertani (LB) broth or on LB agar supplemented as needed with 10 ug/mL chloramphenicol or 150 ug/mL erythromycin. Growth curve experiments were performed in 96-well plate format using a BioTek (Agilent) plate reader set to 37 °C continuous shaking. Starter cultures were grown overnight shaking at 37 °C in BHI with appropriate antibiotics. For the growth curves, the starter cultures were diluted to an OD of 0.01 in fresh BHI in a sterile 96-well plate.

### *sagA* knockout generation

The Δ*sagA::cat* knockout was generated following methods previously described by our laboratory (*Chen et al., 2021*). Briefly, double-stranded DNA (dsDNA) templates were assembled by cloning *sagA* homology arms flanking a chloramphenicol acetyltransferase (*cat*) antibiotic marker into pET21, and amplified by PCR. PCR was performed using Q5 high-fidelity DNA polymerase (New England BioLabs) according to the manufacturer's instructions. The dsDNA was electroporated into electro-competent *E. faecium* Com15 cells harboring RecT prepared using the lysozyme method previously described in *Chen et al., 2021*. The transformants were verified by colony PCR.

*sagA* complementation and mutagenesis *sagA* complementation was accomplished using a pAM401-based plasmid containing *sagA* under the control of its native promoter. Primers used to make the p*sagA* plasmids and empty vector are reported in *Supplementary files 5 and 6*. Plasmids were transformed by electroporation into electrocompetent Δ*sagA* cells.

### Western blot analysis

Cultures were grown overnight shaking at 37 °C with appropriate antibiotics. The next day, the cultures were centrifuged and the pellets were separated from the supernatants. The pellets were resuspended in lysis buffer (50 mM Bis-Tris pH 7.5, 4% SDS, 0.1 mg/mL lysozyme, 25 U benzonase), and then lysed by bead beating (FastPrep system, MP Biomedicals). Proteins were precipitated from the supernatants by methanol-chloroform precipitation and resuspended in water. Proteins from the pellets and supernatants were quantified by BCA analysis (Thermo Fisher), and run on a Stain-Free gel (Bio-Rad). Total protein was visualized by Stain-Free imaging technology using a Bio-Rad ChemiDoc MP imager. Protein bands were then transferred to a nitrocellulose membrane by semi-dry transfer. The membrane was blocked for 1 hr with TBST + 5% powdered milk at room temperature, then stained with primary antibody (rabbit anti-SagA polyclonal sera) diluted 1:50,000 in TBST + 5% milk for 1 hr at room temperature. The membrane was then stained with a secondary antibody (goat anti-rabbit antibody, HRP-conjugate) diluted 1:20,000 in TBST + 5% powdered milk for 1 hr at room temperature with agitation. Membranes were washed with TBST three times for 5 min at room temperature with agitation. Blots were developed using Clarity Western ECL substrate (Bio-Rad) and imaged using a Bio-Rad ChemiDoc MP imager.

### Transmission electron microscopy

Cultures were grown overnight shaking at 37 °C in BHI with appropriate antibiotics. Bacteria were then rinsed with 0.1 M cacodylate buffer followed by immersion in oxygenated 2.5% glutaraldehyde

and 4% paraformaldehyde fixative in 0.1 M sodium cacodylate buffer (pH 7.1) and embedded in low-melting-point agar then fixed overnight at 4 °C. After washing in 0.1 M sodium cacodylate buffer, the samples were post-fixed in buffered 1% osmium tetroxide plus 1.5% potassium ferrocyanide for 1 hr at 4 °C, rinsed in ddH$_2$O and stained *en bloc* with 0.5% uranyl acetate overnight at 4 °C. Samples were washed in ddH$_2$O and dehydrated through a graded ethanol series followed by acetone, and infiltrated with LX-112 (Ladd) epoxy resin and polymerized at 60 °C. Thin sections (70 nm) were imaged at 80kV with a Thermo Fisher 1Talos L120C transmission electron microscope and images were acquired with a CETA 16 M CMOS camera.

## Cryo-electron tomography

*E. faecium* strains used in the cryo-ET experiments were grown overnight at 37 °C in BHI broth with appropriate antibiotics. Fresh cultures were prepared from a 1:100 dilution of the overnight culture and then grown at 37 °C to late log phase. The culture was centrifuged at 1000 x g for 5 min. The pellet was resuspended with BHI broth containing 5% glycerol to OD600 of 3. 5 µl of *ΔsagA* and *ΔsagA*/p*sagA* samples were deposited onto freshly glow-discharged (Pelco easiGlow; 25 s glow at 15mA) Quantifoil R2/1 copper 200 mesh grids for 1 min, back-side blotted with filter paper (Whatman Grade 1 filter paper), and frozen in liquid ethane using a gravity-driven homemade plunger apparatus (inside a 4° cold room with a≥95% relative humidity). The WT Com15 samples were frozen using a Vitrobot Mark IV (Thermo Fisher Scientific) in liquid ethane/propane mixture. The Vitrobot was set to 22 C° at 90% humidity, and manually back-side blotted. The vitrified grids were later clipped with Cryo-FIB autogrids (Thermo Fisher Scientific) prior to milling.

Cryo-FIB milling was performed using Aquilos dual-beam cryo-FIB/SEM instrument (Thermo Fisher Scientific). The vitrified sample was sputtered with metallic platinum for 15 s, followed by a coating of organometallic platinum for 8–9 s to protect the sample, and then the sample was sputtered with metallic platinum for 15 s to prevent drifting during milling. Target sites were milled manually with a gallium ion beam to generate lamellae with a thickness of approximately <150 nm. Finally, the lamellae were sputtered with metallic platinum for 3–4 s to create bead-like fiducial inclusions that aid in tilt series alignment.

Lamellae were imaged with a Titan Krios microscope (Thermo Fisher Scientific) equipped with a field emission gun, an energy filter, and a direct-detection device (Gatan K3). An energy filter with a slit width of 20 eV was used for data acquisition. The SerialEM package (*Mastronarde, 2005*) with PACE-tomo scripts (*Eisenstein et al., 2023*) was used to collect 39 image stacks at a range of tilt angles between +57° and –57° (3° step size) using a dose-symmetric scheme with a cumulative dose of ~117 e$^-$/Å2. Data was collected with a magnification resulting in 2.64 Å/pixel and a nominal defocus of ~ –5 µm. Image stacks containing 10–15 images were motion-corrected using Motioncor2 (*Zheng et al., 2017*) and then assembled into drift-corrected stacks using IMOD. The drift-corrected stacks were aligned and reconstructed by IMOD marker-dependent alignment (*Mastronarde and Held, 2017*). Representative tomograms and raw tilt series are publicly available: EMD-42074, EMD-42086, EMD-42087, and EMPIAR-11692.

All the tomograms were denoised with IsoNet (*Liu et al., 2022*), a deep learning-based software package. Segmentation was performed using Amira software (Thermo Fisher Scientific). All the renderings were visualized in 3D using UCSF ChimeraX (*Goddard et al., 2018*).

### Antibiotic sensitivity assays

MIC test strip analysis. Overnight cultures were diluted to an OD of 0.2. 200 µL of diluted culture were then spread onto a BHI agar plate using glass beads (for strains containing p*sagA* or empty pAM401E, BHI plates with 50 µg/mL erythromycin were used). Plates were placed in a 37 °C incubator for approximately 1 hr to allow the liquid to soak into the plate. MIC test strips (Liofilchem) were then placed at the center of the agar plates using sterile tweezers. Plates were incubated overnight at 37 °C and photographed the next day. For ampicillin liquid culture MIC experiments, the same protocol as the growth curve experiments described above were used, except the cells were treated with various serially-diluted concentrations of ampicillin.

## NOD2 activation assays

The assay was followed by the manufacturer's protocol. In brief, HEK-Blue hNOD2 cells (InvivoGen) were seeded in 96-well plate at the cell density 6 × 10$^4$ cells/well. *E. faecium* were grown to an OD

of ~0.5 from overnight inoculant, washed with PBS, and resuspended in Opti-MEM. Live bacteria were given to HEK-Blue cells at multiplicity of infection (MOI) 1 and incubated for 4 hr at 37 °C. After 4 hr, HEK-Blue cells were carefully washed with PBS and the media was replaced with DMEM supplemented with gentamycin (250 μg/mL) to kill extracellular bacteria. Cells were incubated at 37 °C for an additional 16 hr. NOD2 activation was measured by detecting NF-κB-inducible secreted embryonic alkaline phosphatase expression in the media using colorimetric QUANTI-Blue detection assay (InvivoGen). Fold-change of NOD2 activation was expressed relative to untreated HEK-Blue cells control.

## LC-MS analysis of peptidoglycan

Peptidoglycan was extracted from bacterial sacculi (wild-type *E. faecium*, Δ*sagA*, and Δ*sagA*/p*sagA*) and digested with mutanolysin from *Streptomyces globisporus* (Sigma, 10 KU/ml of mutanolysin in ddH$_2$O) as previously described *Kim et al., 2019*. The resulting soluble muropeptide mixture was treated with sodium borohydride in 0.25 M boric acid (pH 9) for 1 hr at room temperature, quenched with orthophosphoric acid, and pH adjusted to 2–3. Samples were centrifuged at 20,000 × g for 10 min. Then, the reduced peptidoglycan was analyzed by 1290 Infinity II LC/MSD system (Agilent technologies) using Poroshell 120 EC-C18 column (3x150 mm, 2.7 μM). Samples were run at flow rate 0.5 mL/min in mobile phase (A: water, 0.1% formic acid) and an eluent (B: acetonitrile, 0.1% formic acid) using the following gradient: 0–5 min: 2% B, 5–65 min: 2–10% B. The absorbance of the eluting peaks was measured at 205 nm. Masses of peaks were detected with MSD API-ES Scan mode (m/z=200–2500) (*Supplementary file 7*). For quantification of relative abundance of muropeptides, the area under the curve of individual peak from chromatograms was integrated and the percentage of individual peak was calculated relative to all assigned peaks.

## Animals

Specific pathogen-free, seven-week-old male C57BL/6 (B6,000664) mice were obtained from Scripps Rodent Breeding Colony. For breeding, *Nod2$^{-/-}$* (B6.129S1-*Nod2$^{tm1Flv}$*/J) and C57BL/6 (B6,000664) mice were obtained from The Jackson Laboratory. *Nod2$^{+/-}$* and *Nod2$^{-/-}$* littermate cohorts were generated by in-house breeding. Genotyping was performed according to the protocols established for the respective strains by The Jackson Laboratory (Protocol 25069). Mice were housed in autoclaved caging with SaniChip bedding and enrichment for nest building on a +/- hr light/dark cycle. Mice were provided gamma-irradiated chow (LabDiet, 5053) and sterile drinking water ad libitum. Animal care and experiments were conducted in accordance with NIH guidelines and approved by the Institutional Animal Care and Use Committee at Scripps Research (Protocol AUP-21–095).

## Tumor challenge, growth, and treatment experiments

Seven-week-old mice were pre-treated with antibiotic (5 g L-1 streptomycin, 1 g L-1 colistin sulfate, and 1 g L-1 ampicillin) for one week prior to bacterial colonization, as previously described (*Griffin et al., 2021*). *E. faecium* Com15 and Δ*sagA* strains were grown to late logarithmic phase and diluted to OD600=0.46, then diluted 5 x with drinking water to (OD0.46 equal to 10$^8$ CFU ml$^{-1}$). Three days after bacterial colonization (Day 0), mice were subcutaneously implanted with B16F10 melanoma cells (10$^5$ cells per mice) or MC38 tumor cells (3 × 10$^5$ cells per mice) with Matrigel matrix (Corning, 356231). Once the tumors established (around 100 mm$^3$), tumor volume was measured every two days by digital calipers and was calculated as length × width 2 × 0.5, where the width was the smaller of the two measurements. For B16F10 implanted mice, 20 ug anti–PD-L1 (BioXCell, BP0101) were administered to the mice at Day 6, 9, and 12 by intraperitoneal injection in 200 μL antibody buffer solution (BioXCell, PH 6.5 dilution buffer). For MC38 implanted mice, 100 ug anti–PD-1 (BioXCell, BP0146) were administered to the mice at Day 5, 7, 9, 11, and 14 by intraperitoneal injection in 200 μL antibody buffer solution (BioXCell, PH 7.0 dilution buffer). Mice were euthanized with CO2 asphyxiation when the tumors larger than 1.5 cm$^3$ or any ulceration or lesioning developed.

## Fecal colonization analysis

Fecal samples were sterilely collected six days after the start of bacterial administration. Fecal samples were weighed, resuspended in sterile PBS, homogenized by douncing with sterile pestles, serially diluted in sterile PBS, and then plated by drip assay onto selective HiCrome *Enterococcus faecium* agar plates (HIMEDIA 1580) with *Enterococcus faecium* selective supplement (FD226, HIMEDIA).

Plates were incubated for 48 hr at 37 °C under ambient atmosphere and colonies were manually counted.

## Cell isolation and flow cytometry analysis

Tumor dissection and cell isolation were performed as previously described *Griffin et al., 2021*. The dissected tumor samples were placed in RPMI with 1.5 U mL$^{-1}$ Liberase TM (Roche 5401119001), 0.2 mg mL-1 DNase I (Worthington Biochemical LS002006) and ceramic spheres (6.35 mm, MP Biomedicals 116540424-CF) for 30 min at 37 °C with gentle shaking. Samples were then filtered and resuspended in 5 mL of red blood cell lysis buffer (Thermo Fisher 00-4333-57) for 5 min at room temperature. Cells were washed and incubate with 1:1000 Zombie Yellow stain (BioLegend 423103) for 20 min at room temperature. After two times wash, samples were incubated with 20 µL of staining buffer containing 0.5 µL of TruStain FcX anti–mouse CD16/32 blocking agent (BioLegend 101319) for 30 min at room temperature. Sample directly incubate with anti-CD4 (BUV496, GK1.5, BD Biosciences 612953) and anti-PD-1 (BV711, CD279, BD Biosciences 135231) for another 30 min on ice. Cells were washed twice, fixed, and permeabilized with the FoxP3/Transcription Factor Staining Buffer Set (Thermo Fisher 00-5523-00) overnight at 4 °C. Cells were washed twice with perm buffer and then incubated with 20 µL of perm buffer containing 10% rat serum (Thermo Fisher 24-5555-94) for 20 min on ice. Cells were then stainied with the following antibodies for 20 min on ice: anti-CD45 (APC-Fire 750, 30-F11, BioLegend 103153), anti-CD3 (BV785, 17A2, BioLegend 100232), anti-NK1.1 (BV480, PK136, BD Biosciences 746265) anti-CD8 (PE-Cy-7, 53–6.7, BioLegend 100721), anti-FoxP3 (AF532, FJK-16s, Thermo Fisher 58-5773-80), anti-Granzyme B (PE-CF594, GB11, BD Biosciences 562462) and anti-Ki67 (FITC, SolA15, LifeTech 11-5698-82). Samples were analyzed using Cytek Aurora spectral flow cytometer and the data were analyzed using FlowJo Version 10.9.0.

## Acknowledgements

This project was funded by the National Institutes of Health R01 CA245292 grant to HCH and Scripps Research start-up funds to HCH and DP. AM was supported by David C Fairchild Endowed Fellowship to the Skaggs Graduate Program at Scripps Research. This work used equipment supported by NIH grant S10OD032467. We thank Victor Chen for guidance on RecT-mediated recombineering in *E. faecium*. We thank Scott Henderson, Kimberly Vanderpool, and Theresa Fassel for assistance and transmission electron microscopy analysis in the Scripps Microscopy Core.

## Additional information

### Competing interests

Howard C Hang: has filed patent applications (PCT/US2016/028836, PCT/US2020/019038) for the commercial use of SagA-bacteria to improve intestinal immunity and checkpoint blockade immunotherapy, which has been licensed by Rise Therapeutics for probiotic development. The other authors declare that no competing interests exist.

### Funding

| Funder | Grant reference number | Author |
| --- | --- | --- |
| National Institutes of Health | CA245292 | Howard C Hang |
| Scripps Research Institute | Start-up funds | Donghyun Park<br>Howard C Hang |
| Scripps Research Institute | David C. Fairchild Endowed Fellowship | Abeera Mehmood |

The funders had no role in study design, data collection and interpretation, or the decision to submit the work for publication.

## Author contributions
Steven Klupt, Conceptualization, Formal analysis, Investigation, Methodology, Writing – original draft, Conceived the project, Performed microbiology studies, Wrote the manuscript, which was edited by all the other authors; Kyong Tkhe Fam, Formal analysis, Validation, Investigation, Visualization, Methodology, Writing – review and editing, Performed LC-MS analysis of peptidoglycan and NOD2 activation assays, Edited the manuscript, Revised the manuscript; Xing Zhang, Formal analysis, Validation, Investigation, Visualization, Methodology, Writing – review and editing, Performed immune checkpoint inhibitor antitumor and flow cytometry studies, Edited the manuscript; Pavan Kumar Chodisetti, Formal analysis, Writing – review and editing, Assisted microbiology studies; Abeera Mehmood, Formal analysis, Writing – review and editing, Assisted immune checkpoint inhibitor antitumor and flow cytometry studies; Tumara Boyd, Formal analysis, Performed cryo-electron tomography analysis; Danielle Grotjahn, Resources, Writing – review and editing, Established Scripps Research cryo-ET facility, Provided training and technical assistance; Donghyun Park, Formal analysis, Performed cryo-electron tomography analysis; Howard C Hang, Conceptualization, Resources, Data curation, Supervision, Funding acquisition, Methodology, Writing – original draft, Project administration, Writing – review and editing, Conceived the project, Wrote the manuscript, which was edited by all the other authors, Revised the manuscript

## Author ORCIDs
Kyong Tkhe Fam (iD) https://orcid.org/0000-0002-4877-5679
Howard C Hang (iD) https://orcid.org/0000-0003-4053-5547

## Ethics
Animal care and experiments were conducted in accordance with NIH guidelines and approved by the Institutional Animal Care and Use Committee at Scripps Research (Protocol AUP-21-095).

Reviewer #1 (Public Review): https://doi.org/10.7554/eLife.95297.3.sa1
Author response https://doi.org/10.7554/eLife.95297.3.sa2

---

# Additional files

## Supplementary files
- Supplementary file 1. Mutations detected in Δ*sagA E. faecium* Com15 strain.
- Supplementary file 2. Summary of MIC determinations via antibiotic test strips for *E. faecium* wild-type (WT), Δ*sagA,* and Δ*sagA/*p*sagA* (*Figure 1—figure supplement 2a*).
- Supplementary file 3. *E. faecium* strains used in this study.
- Supplementary file 4. Plasmids used in this study.
- Supplementary file 5. Primers used in this study for generating complementation plasmids and empty vector.
- Supplementary file 6. Primers used in this study for SagA mutagenesis.
- Supplementary file 7. Masses of peptidoglycan fragments were detected with MSD API-ES.
- MDAR checklist

## Data availability
All data generated within this study are available in the article. Cryo-ET raw data and representative tomograms have been deposited to the Electron Microscopy Public Image Archive under accession number EMPIAR-11692 and Electron Microscopy Data Bank under accession numbers EMD-42074, EMD-42086, and EMD-42087.

The following datasets were generated:

| Author(s) | Year | Dataset title | Dataset URL | Database and Identifier |
|---|---|---|---|---|
| Klupt S, Fam KT, Zhang X, Chodisetti PK, Mehmood A, Boyd T, Grotjahn D, Park D, Hang HC | 2024 | Cryo-electron tomography of Enterococcus faecium | https://www.ebi.ac.uk/empiar/EMPIAR-11692 | Electron Microscopy Public Image Archive, EMPIAR-11692 |
| Hang HC, Park D | 2024 | Representative tomogram of Enterococcus faecium WT Com15 | https://www.ebi.ac.uk/emdb/EMD-42074 | EMDataBank, EMD-42074 |
| Hang HC, Park D | 2024 | Representative tomogram of Enterococcus faecium SagA complementation strain | https://www.ebi.ac.uk/emdb/EMD-42086 | EMDataBank, EMD-42086 |
| Hang HC, Park D | 2024 | Representative tomogram of Enterococcus faecium SagA deletion strain | https://www.ebi.ac.uk/emdb/EMD-42087 | EMDataBank, EMD-42087 |

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
