## [Editor Report · eLife assessment]

The authors build upon prior data implicating the secreted peptidoglycan hydrolase SagA produced by *Enterococcus faecium* in immunotherapy. Leveraging new strains with sagA deletion/complementation constructs, the investigators reveal that sagA is non-essential, with sagA deletion leading to a marked growth defect due to impaired cell division, and sagA being necessary for the immunogenic and anti-tumor effects of *E. faecium*. In aggregate, the study utilizes **compelling** methods to provide both **fundamental** new insights into *E. faecium* biology and host interactions and a proof-of-concept for identifying the bacterial effectors of immunotherapy response.

---

## [Referee Report · Reviewer #1 (Public Review)]

Klupt, Fam, Zhang, Hang and colleagues present a novel study examining the function of sagA in E. faecium, including impacts on growth, peptidoglycan cleavage, cell separation, antibiotic sensitivity, NOD2 activation and modulation of cancer immunotherapy. This manuscript represents a substantial advance over their prior work, where they found that sagA-expressing strains (including naturally-expressing strains and versions of non-expressing strains forced to overexpress sagA) were superior in activating NOD2 and improving cancer immunotherapy. Prior to the current study, an examination of sagA mutant E. faecium was not possible and sagA was thought to be an essential gene.

The study is overall very carefully performed with appropriate controls and experimental checks, including confirmation of similar densities of ΔsagA throughout. Results are overall interpreted cautiously and appropriately.

---

## [Author Response]

The following is the authors’ response to the original reviews.

**eLife assessment**
The authors build upon prior data implicating the secreted peptidoglycan hydrolase SagA produced by Enterococcus faecium in immunotherapy. Leveraging new strains with sagA deletion/complementation constructs, the investigators reveal that sagA is non-essential, with sagA deletion leading to a marked growth defect due to impaired cell division, and sagA being necessary for the immunogenic and anti-tumor effects of E. faecium. In aggregate, the study utilizes compelling methods to provide both fundamental new insights into E. faecium biology and host interactions and a proof-of-concept for identifying the bacterial effectors of immunotherapy response.

We thank the Reviewers for their positive feedback on our manuscript. We also appreciate their helpful comments/critiques and have revised the manuscript as indicated below.

**Public Reviews:**

**Reviewer #1 (Public Review):**
Klupt, Fam, Zhang, Hang, and colleagues present a novel study examining the function of sagA in E. faecium, including impacts on growth, peptidoglycan cleavage, cell separation, antibiotic sensitivity, NOD2 activation, and modulation of cancer immunotherapy. This manuscript represents a substantial advance over their prior work, where they found that sagA-expressing strains (including naturally-expressing strains and versions of non-expressing strains forced to overexpress sagA) were superior in activating NOD2 and improving cancer immunotherapy. Prior to the current study, an examination of sagA mutant E. faecium was not possible and sagA was thought to be an essential gene.The study is overall very carefully performed with appropriate controls and experimental checks, including confirmation of similar densities of ΔsagA throughout. Results are overall interpreted cautiously and appropriately.I have only two comments that I think addressing would strengthen what is already an excellent manuscript.In the experiments depicted in Figure 3, the authors should clarify the quantification of peptidoglycans from cellular material vs supernatants. It should also be clarified whether the sagA need to be expressed endogenously within E. faecium, and whether ambient endopeptidases (perhaps expressed by other nearby bacteria or recombinant enzymes added) can enzymatically work on ΔsagA cell wall products to produce NOD2 ligands?

We mentioned in the main text that peptidoglycan was isolated from bacterial sacculi and digested with mutanolysin for LC-MS analysis. We have now also included “mutanolysin-digested” sacculi in the Figure 3 legend as well.

We have added the following text “We next evaluated live bacterial cultures with mammalian cells to determine their ability to activate the peptidoglycan pattern recognition receptor NOD2” and “our analysis of these bacterial strains” to indicate live cultures were evaluated for NOD2 activation.

We have also added the following text “Our results also demonstrated that while many enzymes are required for the biosynthesis and remodeling of peptidoglycan in E. faecium, SagA is essential for generating NOD2 activating muropeptides ex vivo.”

In the murine experiments depicted in Figure 4, because the bacterial intervention is being performed continuously in the drinking water, the investigators have not distinguished between colonization vs continuous oral dosing of the mice peptidoglycans. While I do not think additional experimentation is required to distinguish the individual contributions of these 2 components in their therapeutic intervention, I do think the interpretation of their results should include this perspective.

We have added the following text “We note that by continuous oral administration in the drinking water, live E. faecium and soluble muropeptides that are released into the media during bacterial growth may both contribute to NOD2 activation in vivo.” and revised the following text “Nonetheless, these results demonstrate SagA is not essential for E. faecium colonization, but required for promoting the ICI antitumor activity through NOD2 in vivo.

**Reviewer #2 (Public Review):**
Summary:The gut microbiome contributes to variation in the efficacy of immune checkpoint blockade in cancer therapy; however, the mechanisms responsible remain unclear. Klupt et al. build upon prior data implicating the secreted peptidoglycan hydrolase SagA produced by Enterococcus faecium in immunotherapy, leveraging novel strains with sagA deleted and complemented. They find that sagA is non-essential, but sagA deletion leads to a marked growth defect due to impaired cell division. Furthermore, sagA is necessary for the immunogenic and anti-tumor effects of E. faecium. Together, this study utilizes compelling methods to provide fundamental new insights into E. faecium biology and host interactions, and a proof-of-concept for identifying the bacterial effectors of immunotherapy response.Strengths:Klupt et al. provide a well-written manuscript with clear and compelling main and supplemental figures. The methods used are state-of-the-art, including various imaging modalities, bacterial genetics, mass spectrometry, sequencing, flow cytometry, and mouse models of immunotherapy response. Overall, the data supports the conclusions, which are a valuable addition to the literature.Weaknesses:Only minor revision recommendations were noted.
**Recommendations for the authors:**

**Reviewer #2 (Recommendations For The Authors):**
General comments - the number/type of replicates and statistics are missing from some of the figure panels. Please be sure to add these throughout - all main figure panels should have replicates. I've also noted some specific cases below.Abstract - sagA is non-essential, need to edit text at "essential functions".

This change has been made.

"small number of mutations" - specify how many in the text.

We revised the text. “Small number” is changed to “11”.

"under control of its native promoter" - what was the plasmid copy number? It looks clearly overexpressed in Figure 1d despite using a native promoter, although it's a bit hard to know for sure without a loading control.

pAM401 has p15A origin of replication, therefore the plasmid copy number ~20-30 copies (Lutz R. et al Nucleic Acids Res. 1997). Total protein was visualized by Stain-Free imaging technology (BioRad) and serves as protein loading control and has been relabeled accordingly.

"decrease levels of small muropeptides" - the asterisks are missing from Figure 3a.

Green asterisks for peaks 2, 3, 7 and purple asterisks for peaks 13, 14 were added.

The use of "Com 15 WT" in the figures is confusing - just replace it with "wt" and specify the strain in the text. Presumably, all of the strains are on the Com 15 background.

“Com15 WT” was replaced to “WT” in figures and main text.

Change 1d to 1b so that the panels are in order (reading left to right and then top to bottom).Figure 1 legend is missing a number of replicates and statistics for 1a.

Number of replicates were added.

Figure 1b - it's unclear to me what to look at here, could add arrows indicating the feature or interest and expand the relevant text.

Arrows pointing to cell clusters were added.

Figure 1d - what is "stain free"? It would be preferable to show a loading control using an antibody against a constitutive protein to allow for normalization of the loading control.

Stain-Free Imaging technology (BioRad) utilizes gel-containing trihalo compound to make proteins fluorescent directly in the gel with a short photoactivation, allowing the immediate visualization of proteins at any point during electrophoresis and western blotting. Stain-Free total protein measurement serves as a reliable loading control comparable to Coomassie Blue Staining. This has been relabeled a “Total protein” in the Figure and Stain-free imaging technology is noted in the legend.

ED Figure 1 - representative of how many biological replicates?

Legends are updated.

ED Figure 2a - I would replace this with a table, it's not necessary to show the strip images. Also, please specify the number of replicates per group.

Additional Extended Data Table 2 was added.

ED Figure 2b - This data was not that convincing since the sagA KO has a marked growth defect and the time points are cut off too soon to know if growth would occur later. The MIC definition is potentially misleading. Should specific a % growth cutoff (i.e. <10% of vehicle control) and the metric used (carrying capacity or AUC). Then assign MIC to the tested concentration, not a range. The empty vector also seems to impact MIC, which is concerning and complicates the interpretation. Specify the number of replicates and add statistics. Given these various concerns, I might suggest removing this figure, as it doesn't really add much to the story.

We appreciate this comment from the Reviewer, but believe this data is helpful for paper and have included longer time points for the growth data. The definition of MIC for ED Fig. 2b has been included in the legend.

Figure 2 - specify the type of replicate. Number of cells? Number of slices? Number of independent cultures?

For Cryo-ET experiments single bacterial cultures were prepared. Number of cells and slices for analysis are indicated in the legend. Legends are updated.

Figure 4e - missing the water group, was it measured?

Water (αPD-L1) group was not included in immune profiling of tumor infiltrating lymphocytes (TILs) experiment, as we have previously demonstrated limited impact on ICI anti-tumor activity and T cell activation in this setting (Griffin M et al Science 2021).

Figure 4d - is this media specific to your strains? If not, qPCR may be a better method using strain-specific primers.

Yes, HiCrome Enterococcus faecium agar plates (HIMEDIA 1580) are selective for Enterococcus species, moreover the agar is chromogenic allowing to identify E. faecium as yellow colonies among other Enterococcus species.